# The Role of Immersive Experience in Anxiety Reduction: Evidence from Virtual Reality Sessions

**DOI:** 10.3390/brainsci15010014

**Published:** 2024-12-26

**Authors:** Dominika Wilczyńska, Tamara Walczak-Kozłowska, David Alarcón, María José Arenilla, Jose Carlos Jaenes, Marcelina Hejła, Mariusz Lipowski, Joanna Nestorowicz, Henryk Olszewski

**Affiliations:** 1Faculty of Social and Humanities, University WSB Merito, 80-266 Gdańsk, Poland; 2Department of Neuropsychology, Institute of Psychology, University of Gdansk, 80-309 Gdańsk, Poland; tamara.walczak@ug.edu.pl; 3Social Anthropology, Basic Psychology and Public Health Department, Pablo de Olavide University, 41013 Sevilla, Spain; dalarub@upo.es (D.A.); mjarevil@alu.upo.es (M.J.A.); jcjaesan@upo.es (J.C.J.); 4Department of Physical Culture, Gdansk University of Physical Education and Sport, 80-336 Gdańsk, Poland; marcelinahejla@gmail.com; 5Institute of Polish Herbal Medicine and Natural Therapies, 97-360 Kamieńsk, Poland; science@instytutzielarstwa.com (M.L.); biuro@instytutzielarstwa.pl (J.N.); 6Department of Psychology and Developmental Psychopathology, University of Gdansk, 80-309 Gdańsk, Poland; henryk.olszewski@ug.edu.pl

**Keywords:** virtual reality, immersion, emotional state, wheelchair condition

## Abstract

Background/Objectives: Virtual reality (VR) is an innovative technology with the potential to transform digital experiences, particularly in relation to mental health concerns such as anxiety. Therefore, this study investigates the potential of a newly designed VR experience to alleviate anxiety by focusing on the mediating role of VR-induced immersion. Methods: The study included 419 individuals aged 10 to 80 years, with 29 aged 10–15 years and 390 above 15 years, who were randomly assigned to experimental and control groups on the basis of project-defined criteria, including a random allocation to the wheelchair-using group. Both groups used goggles for virtual space navigation, with the experimental group employing a multijoint arm mounted on an aluminum frame and special algorithms to navigate without controllers. We assessed immersion in VR using the Polish adaptation of the Immersion Questionnaire and anxiety using the Polish adaptations of the State–Trait Anxiety Inventory (STAI-X1) and its early adolescent version, the State–Trait Anxiety Inventory–Children (STAI-C1). Results and Conclusions: The results indicate that individuals using the new VR device demonstrated increased immersion and reduced post-test anxiety levels, highlighting the significance of immersion in enhancing positive affect, mitigating the negative effects of VR technology, and offering insights for future development and refinement of VR solutions.

## 1. Introduction

Virtual reality (VR) is a computer-generated, immersive simulation of a three-dimensional (height, width, and depth) environment that allows individuals to interact with it in a way that feels remarkably lifelike, achieved through the use of specialized electronic equipment [1]. This heightened sense of realism gives rise to a phenomenon called “presence”, signifying the user’s genuine feeling of inhabiting the virtual world of computer-generated visual and auditory stimuli. Owing to its ability to provide users with highly realistic immersive experiences, VR has emerged as a powerful tool capable of creating novel consumption experiences [2]. Consequently, it has found applications across diverse sectors, including gaming, education, and health care [1]. However, one reason VR is so powerful in psychological research is that it offers experimental control similar to conventional lab settings while providing interactive, dynamic, and realistic stimuli and environments. This is accomplished through VR’s technical capabilities, facilitating a deep dive into real-life experiences, a concept commonly termed “immersion” [3,4,5].

Steuer [6] defined immersion as the degree to which computer displays create a comprehensive, expansive, enveloping, and lifelike illusion of reality for human senses. It encompasses inclusiveness, accommodating multiple sensory modalities, i.e., expansiveness and panoramic experience, surroundings, richness, resolution, and quality of sensory information, i.e., vividness. Immersion is very important and, according to some researchers studying the virtual reality of computer games, is crucial for the satisfaction derived from experiencing augmented reality [7]. For example, in recent research by Christou [8], a strong correlation between immersion and the evaluation of game attractiveness was identified. This phenomenon persisted irrespective of the game’s theme or the players’ proficiency level, suggesting an inherent link between game engagement and its assessment. Nevertheless, Christou observed a lack of definitive results in determining the direction of the correlation between game satisfaction and immersion. Building on these insights into the significance of immersion, more recent research on immersive experiences in virtual reality (VR), both with and without controllers, reveals varying effects on the user experience, immersion, and interaction capabilities. While joysticks offer ease of use, leaning-based interfaces and algorithm-driven systems provide more immersive and engaging experiences in VR. These systems particularly reduce motion sickness and cognitive load [9] while enhancing the sense of presence by enabling a more interactive and enjoyable experience through physical and natural movement [10,11]. The theoretical foundation for these findings can be drawn from Embodied Cognition (EC) theories, which posit that human cognition is deeply rooted in the body’s interactions with its environment [12]. EC suggests that cognitive processes are fundamentally shaped by the sensorimotor capabilities of an individual situated in a complex, real-world environment. From this perspective, the body not only supports but actively constrains and structures the mind, where perception and cognition primarily serve action in real-time. Cognition thus emerges from the dynamic interaction between the organism and its surroundings [13]. Based on these principles, we hypothesize that modern VR, by offering more immersive and realistic body–environment interactions, can foster deeper cognitive and emotional experiences. In contrast, classical VR may not fully evoke the embodied experience to the same extent, leading to different psychological and behavioral outcomes. Therefore, we predict that modern VR, through better simulation of physical movements, will yield stronger positive emotional responses compared to classical VR. In the realm of technology and entertainment, virtual reality (VR) has emerged as a groundbreaking innovation that has the potential to revolutionize the way we experience and interact with digital environments. Although VR holds immense promise, it also raises some questions, notably regarding its effects on mental health. Among the many dimensions of mental health that have been explored related to VR, anxiety has garnered significant attention. Amidst this landscape, the rise of VR technology has prompted both excitement and concern regarding its potential influence on anxiety levels. On the one hand, VR offers a powerful tool for exposure therapy, a well-established psychological treatment for anxiety disorders. On the other hand, the immersive nature of VR experiences has raised questions about whether they inadvertently exacerbate anxiety in certain contexts. Immersive VR environments that closely replicate reality can intensify anxiety, especially in scenarios involving negative or threatening situations. For example, a study involving fire safety VR training revealed that when the virtual environment reproduced reality more closely, users experienced greater anxiety; however, this also increased their engagement and learning effectiveness [14]. A study conducted by Lavoie et al. [15] investigated the potential negative emotional consequences of VR gameplay. Research has revealed that specific VR scenarios can evoke profoundly negative emotional responses. Compared with those using laptops, participants engaging in an interactive VR scenario reported heightened levels of absorption, consequently intensifying their negative emotional reactions to the scenario. A follow-up questionnaire conducted several hours later revealed a positive correlation between the augmented negative emotions experienced in VR and negative rumination—namely, detrimental self-related thoughts associated with distress. Additionally, a study by Lemmens et al. [16] revealed that playing games in VR resulted in a stronger sense of presence, lower heart rate variability, and a stronger subjective sense of fear than playing on a TV. The feeling of presence thereby mediated the effects of VR on fear. Both studies revealed that VR environments can potentially elicit profoundly negative emotional responses, posing a risk to users if not effectively controlled. On the other hand, a study by Li et al. [17] investigated the effects of VR-based exposure therapy on anxiety and depression. The study revealed that VR-based exposure therapy was effective in reducing anxiety and depression symptoms in patients with anxiety disorders. Additionally, Freeman et al. [18] demonstrated that VR can be highly effective in treating anxiety disorders, especially through exposure-based therapy. Immersive VR environments allow individuals to confront anxiety-provoking stimuli in a controlled manner. The presence felt in VR heightens the sense of being in a real situation, which helps elicit psychophysiological anxiety responses similar to those encountered in real life, making the therapy more effective. A review by Marín-Morales et al. [19] investigated the use of VR for emotion recognition research. The authors reported that immersive VR environments can elicit emotional responses with high levels of sense of presence and interactivity, which can be useful for studying human emotions. Shafer et al. [20] revealed that playing VR games with higher technological quality did not reduce cybersickness, but sensory conflict had a significant effect on how sick users became, and enjoyment of VR games was directly influenced by a sense of spatial presence, which was affected by interactivity and realism. These preliminary findings suggest that VR may elicit differential cognitive and social responses compared with less immersive technology [21]. However, the authors emphasize that further research is needed to better understand the effects of immersion on emotions and mental health [19].

### 1.1. Aim of the Study

Considering the established body of knowledge and research mentioned earlier and recognizing the ongoing need for further exploration into the relationship between virtual reality (VR) and emotions, our goal was to evaluate the potential of various VR solutions (both conventional and contemporary, offering improved control over movements) to enhance the immersive experience. Additionally, we aimed to closely examine the connection between the level of virtual reality immersion and the user’s immediate emotional response. Our objective was to analyze the impact of two different VR technologies—traditional ones commonly used (headsets with controllers) and modern ones (headsets attached to a multijoint arm without the need for controllers)—on the intensity of respondents’ anxiety while also considering the role of VR immersion in this context. Moreover, we took into account age, sex, and wheelchair condition as control variables [22]. The inclusion of wheelchairs in this study aimed to create a more comprehensive understanding of the VR experience across diverse mobility conditions. Given that VR experiences can vary significantly based on physical mobility, we wanted to evaluate whether the new VR device, which uses a multijoint arm for navigation, could offer a similarly immersive and anxiety-reducing experience for both individuals using wheelchairs and those not using them. This consideration was particularly important because mobility constraints can influence the perception of immersion and anxiety levels in VR environments [23]. Moreover, this study is part of a project aimed at implementing new technology in museums to provide individuals using wheelchairs the opportunity to explore history, even if they cannot physically visit the museum. To the best of our knowledge, this study represents the first comparison between modern VR solutions and traditional VR technologies in terms of their ability to enhance the virtual reality experience (inducing immersion in the user). Moreover, there is currently no research evaluating the potential of newly designed VR solutions in alleviating negative mental states. This study aims to address this gap in existing research, thereby serving as a crucial step toward exploring and applying modern VR solutions [22].

### 1.2. Study Design

This study employed a between-subjects experimental design to investigate the effects of two key variables on immersion and anxiety during virtual reality (VR) experiences. The study manipulated two independent variables: the level of VR technology and mobility condition, resulting in a 2 × 2 factorial design. Participants were randomly assigned to one of the four experimental conditions: (1) traditional VR + without wheelchair, (2) modern VR + without wheelchair, (3) traditional VR + with wheelchair, or (4) modern VR + with wheelchair. Each participant experienced only one combination of conditions, ensuring that both independent variables operated as between-subjects factors.

### 1.3. Research Questions and Hypotheses

This study explores the following research questions, along with the corresponding hypotheses:
Does the use of modern VR technology (with a multijoint arm and natural navigation) result in higher levels of immersion compared to traditional VR technology (with controllers)?
**H1.** *Participants using modern VR technology will report significantly higher immersion levels compared to those using traditional VR technology.*
2.How does the type of VR technology influence changes in anxiety levels before and after the VR session?
**H2.** *Modern VR technology will lead to greater reductions in post-session anxiety compared to traditional VR technology.*
3.Do mobility conditions (with or without wheelchair) affect immersion levels and anxiety reduction in VR environments?
**H3.** *Mobility condition (with or without wheelchair) will not significantly impact immersion levels or anxiety reduction.*
4.Does immersion mediate the relationship between VR technology (group) and anxiety levels after the VR session (T1), when controlling for variables such as sex, age, baseline anxiety (T0), and mobility condition?
**H4.** *Immersion will act as a mediator between VR technology (group) and post-session anxiety (T1), such that higher immersion levels associated with modern VR technology will explain greater reductions in anxiety.*

## 2. Materials and Methods

### 2.1. Participants

The study group consisted of 419 individuals aged between 10 and 80 years (13.37 ± 1.71; M ± SD). There were 29 individuals aged 10–15 years (13.37 ± 1.71; M ± SD) and 390 individuals above the age of 15 (34.68 ± 15.71; M ± SD) who were randomly assigned to either the experimental or the control group according to the initial assumptions defined for the specific project phase, as described in the respective stage. The allocation to the wheelchair-using group was also random. No significant differences were found between the research conditions (control vs. experimental) for sex (X^2^ = 2.27, *p* = 0.13), wheelchair use (X^2^ = 0.40, *p* = 0.53), or age (*t* = 0.46, *p* = 0.23). Detailed descriptive statistics for each subgroup are presented in Table 1.

### 2.2. Materials

#### 2.2.1. Virtual Reality (VR) Technological Solutions

Test environments were prepared for the experimental and control groups via the Unreal Engine 4.22 engine. In both cases, the participants navigated in a virtual reality (VR) space using Oculus Quest 2 goggles. The VR scenario involved exploring a room furnished with pre-World War II furniture. The scenario was designed to be neutral, eliciting no emotional response from the participants. The goggles for the experimental group were attached to a multijoint arm mounted on an easily assembled, relatively lightweight aluminum frame. The experimental group was assessed in a test environment using specially developed algorithms for navigating the virtual space without controllers. The participants moved naturally within the real-world space up to the designated boundary, and the multijoint arm on the aluminum frame tracked their position. Once the boundary was reached, movement in the virtual space was controlled through head movements, changes in body rotation, and tilts. Upon returning to the study area, the participants’ movements were again detected from their real-world position. See Figure 1.

The control group was assessed in a test environment without using algorithms, where movement was accomplished using controllers. The participants stood or sat still and navigated within the virtual space using a joystick on the controller. See Figure 2.

#### 2.2.2. Wheelchair Condition

In this study, we deliberately involved healthy individuals using wheelchairs to simulate the experience of disability. This approach allowed us to test the universality of the VR design in a controlled and ethical manner while minimizing variability from other factors such as pre-existing conditions or comorbidities. The wheelchairs used in this study were standard manual wheelchairs. See Figure 3.

#### 2.2.3. Psychological Measurement

The State–Trait Anxiety Inventory, version X1 (STAI-X1), and its children’s version, the STAI-C1 [24,25] assess transient, situational anxiety using 20 statements (e.g., I am calm; I feel anxious). Adults rate their feelings from “definitely not” to “definitely yes” on a four-point scale, with scores ranging from 20 to 80. The children’s version uses a three-point scale, with scores ranging from 20 to 60 (e.g., I feel cheerful; I am confused). Both have been adapted into Polish by Sosnowski et al. [26] for the STAI-X1 and Jaworowska for the STAI-C1 [27]. Cronbach’s alpha for STAI-C1 was 0.84, while for STAI-X1 in the experimental group, it was 0.91, and it was 0.93 in the control group.

The Immersion Questionnaire by Jennett et al. [28], adapted by Strojny and Strojny [29] measures engagement and absorption in video gaming via 27 statements on a 5-point scale, with scores ranging from 27 to 135 (e.g., To what extent were you aware of your presence in the real environment?). The scale demonstrated satisfactory reliability, with Cronbach’s alpha values of 0.79 for the youth group and 0.88 for the experimental and control groups.

### 2.3. Procedure

The study was conducted between 1 October 2022 and 23 December 2022. Participants were recruited through email invitations to schools, universities, and cultural centers in the Tricity area (northern voivodeship of Poland) and through paper notices placed in schools, universities, churches, and cultural centers in Tricity and its surroundings. Exclusion from the study included photosensitivity, epilepsy, poor health condition on the day of the study, and lack of informed consent from the participant or guardian. Those interested in participating in the study contacted Chronospace Company in Sopot, and scheduled a specific day (Monday–Sunday) and time (between 9:00 AM and 5:00 PM). The research was conducted under two conditions, experimental and control, at the following locations in Poland: Chronospace in Sopot, Ateneum University, Academy of Physical Education and Sport, and the Baltic Opera in Gdansk. Upon arriving at the research location, participants completed informed consent forms. In the case of minors, parents signed the informed consent forms, or nonminors provided informed consent forms from their parents. The participants were subsequently randomly assigned to either the experimental or the control group, whose VR goggles were tested. The participants also randomly selected whether they would perform the study in a wheelchair or not. Random allocation to the wheelchair or no-wheelchair condition was determined using a computer-generated sequence to minimize bias and ensure equal representation [22]. The research procedure is presented in Figure 1.

Prior to the study, all participants completed an online anxiety questionnaire, the STAI-C1 (version for children and adolescents up to 15 years old) or the STAI-X1 (version for those aged 16 and older). The participants were then subjected to experimental or control conditions. In both cases, the participants navigated in a virtual space (VR) via goggles. The experimental group moved with goggles attached to a multijoint arm mounted on an aluminum frame. The experimental group was assessed in a test environment using specially developed algorithms for navigating in the virtual space without the use of controllers. The participants moved naturally within the real-world space up to the designated boundary, and the position of the multijoint arm mounted on the aluminum frame was tracked. Once the boundary was reached, movement in the virtual space was controlled through head movements, changes in body rotation, and tilts. Upon returning to the study area, the participants’ movements were again detected from their real-world position. On the other hand, the control group was assessed in a test environment without using algorithms, where movement was accomplished via controllers. The participants stood or sat still, and navigation within the virtual space was performed via a joystick on the controller. The entire VR study, both in the control and experimental groups, lasted 5 min (in a few exceptional cases, the duration of the study in the control condition was shorter because the participants experienced dizziness). The participants subsequently filled out the relevant anxiety questionnaire again: the STAI-C1 or STAI-X1 and the immersion questionnaire. The procedure was the same for participants in wheelchairs [22]. Both groups, experimental and control, navigated the same VR environment, which depicted an old room from before World War II. The room was furnished with antique furniture and paintings, resembling a museum or a historical setting. The experimental and control conditions were conducted indoors at Chronospace in Sopot, the Academy of Physical Education and Sport in Gdansk, the Baltic Opera, and Ateneum University in Gdansk, Poland. This study was conducted following the Declaration of Helsinki and approved by the Ethics Board for Research Projects at the Institute of Psychology, University of Gdansk, the Ethics Board Opinion, in response to inquiry no. 33/2020.

### 2.4. Statistical Analysis

Statistical analyses were performed via IBM SPSS Statistics 20. To determine whether there were differences according to the type of research condition, Pearson’s chi-square test for categorical variables (sex, wheelchair) and a *t* test for continuous variables (age) were used. Normality was not observed, and nonparametric Wilcoxon–Mann–Whitney tests were conducted by type of research condition and for the difference in means between men and women in immersion and anxiety. Spearman’s correlation coefficients of quantitative variables (age, immersion, anxiety at T0, anxiety at T1) were calculated. A regression analysis was conducted to predict anxiety at T1 (experimental/control group, sex, age, wheelchair, anxiety at T0, immersion). The fourth mediation chain model was subsequently examined via PROCESS macro software v4.0 [30].

## 3. Results

### 3.1. Descriptive Statistics

A total of 390 participants were included. The immersion and anxiety scores (means and SDs) of the participants are presented in Table 2. No significant differences were found between the control and experimental conditions for anxiety at T0 (Wm = 40.32, *p* = 0.63). However, significant differences were found in the immersion scores between the experimental and control conditions (Wm = 43.51, *p* < 0.05), with the experimental group showing greater immersion. Significant differences were also found in anxiety at T1 (Wm = 36.37, *p* < 0.001), where the experimental condition had lower anxiety scores than the control condition. Additionally, significant differences by sex were observed for anxiety at T1 within the control group (Wm = 9.47, *p* < 0.01) and for immersion within the experimental group (Wm = 13.29, *p* < 0.01).

### 3.2. Spearman’s Correlation Coefficients of Quantitative Variables

Spearman’s correlation coefficients revealed significant relationships between age and immersion (*r* = 0.22, *p* < 0.01). Moreover, anxiety at T1 was negatively correlated with immersion (r = −0.22, *p* < 0.01) and positively correlated with anxiety at T0 (r = 0.65, *p* < 0.01) (see Table 3).

A linear regression model was constructed to identify the factors predicting participants’ anxiety at T1. The model accounted for 42% of the variance in anxiety at T1 (R^2^ = 0.42, adjusted R^2^ = 0.41, F(6, 383) = 45.55, *p* < 0.001; see Table 4). When controlling for the effects of sex, age, wheelchair use, and baseline anxiety, being in the experimental group (β = −0.18, *p* < 0.001) and higher levels of immersion (β = −0.19, *p* < 0.001) were associated with a reduction in anxiety at T1. In contrast, age (β = 0.12, *p* < 0.01) and baseline anxiety (Anxiety T0; β = 0.54, *p* < 0.001) were linked to an increase in anxiety at T1. In contrast, neither sex (β = 0.06, *p* = 0.17) nor wheelchair use (β = 0.03, *p* = 0.45) significantly predicted anxiety at T1.

The experimental/control group was established as the independent/predictor variable, anxiety at T1 was the dependent/criterion variable, and immersion was the mediating variable. Additionally, sex, age, wheelchair use and anxiety status at T0 were entered as covariates. All of them were included in the mediation model (see Table 5 and Figure 2). The indirect effect of the experimental/control group on anxiety at T1 through immersion is negative and significantly different from zero (point estimate = −0.41, with a 95% bootstrap confidence interval of −0.90 to −0.07). The direct effect is statistically significant according to commonly used standards (c′ = −3.26, *p* < 0.001).

## 4. Discussion

This study investigated the potential role of the newly designed upgraded VR experience in diminishing the negative experience of anxiety. We were particularly interested in the mediating effect of immersion induced by VR experience on the severity of anxiety symptoms. Overall, our results showed that those people who participated in VR sessions with the new (tested) device manifested greater immersion and lower severity of anxiety in the posttest (no group differences were observed in the case of initial anxiety symptoms). Notably, in the experimental group, women experienced greater immersion than men, while in the control group, women showed higher anxiety symptoms in both assessments, highlighting potential gender differences in the response to VR-induced immersion and anxiety. Moreover, the severity of anxiety symptoms measured after VR experience was significantly associated with initial (preexisting) anxiety symptoms, age and immersion. We found that such a negative mental state after the VR experience was perceived more by participants in the group with standard VR equipment. Interestingly, neither sex nor wheelchair condition predicted the severity of anxiety symptoms measured after the VR experience. Taken together, these variables accounted for considerable variance (42%) in the post-VR severity of anxiety. Furthermore, we observed a partial mediating effect with group as a predictor variable, post-VR severity of anxiety as a criterion variable and immersion as a mediator (controlling for sex, age, initial anxiety and wheelchair use).

A lower severity of anxiety measured immediately after the VR experience was observed among the participants in the experimental group (in comparison to the control group), those who were less anxious at baseline, and those in whom we observed greater immersion during the VR experience. We conclude that lower anxiety among people in the experimental group, in comparison with the control group, results from the newly designed VR device that was used, its features and the effects it produces (e.g., greater immersion discussed above). Previous studies have indicated that augmented reality promotes well-being among different cohorts [31,32,33,34,35]. The results of a recent meta-analysis [36] revealed that interventions using VR systems enhance well-being in both healthy and clinical adults and older adults without evident psychopathological conditions. The authors highlighted that there is a growing trend in using technology to promote emotional, psychological and social well-being. These solutions have become more favorable in terms of treatment intensity and duration, cost, and suitability for continuity of care [37,38].

Although we cannot indicate causal relationships in this study, we suppose that using more immersive VR solutions somehow contributes to a lower experience of psychological distress. As mentioned earlier, this VR solution enables increasingly smooth navigation in the virtual space without the use of controllers. McMahan [39], in relation to gaming, noted that for immersion to occur, what happens in the game must meet the user’s expectations and enable actions that are significant for the individual. It is natural for a person who expects freedom of exploration at least at the same level as in reality when faced with augmented reality. The comfort and naturality that this newly developed VR device provides probably enhances immersion more than standard solutions do, which contributes to subjective well-being during use. Moreover, a more immersive VR experience may foster a sense of environmental mastery and a feeling of personal growth and autonomy [36], which has great potential to improve one’s mental state. Moreover, the results of the current study showed that the use of wheelchairs did not diminish the level of immersion or the effectiveness of the VR experience in reducing anxiety. Participants using wheelchairs reported similar levels of immersion and anxiety reduction compared to those not using wheelchairs, suggesting that the new navigation method is effective across varying mobility conditions. This finding is significant as it demonstrates the device’s potential to provide accessible VR experiences for individuals with different physical capabilities. Building on this, we believe that future research should further explore how factors such as mobility, accessibility, and inclusivity in VR environments directly influence mental states like anxiety and emotional well-being, particularly in populations with specific physical or psychological vulnerabilities.

Previous studies have demonstrated that immersive virtual reality has the potential to be a powerful pain control technique, enabling the management and modulation of pain in healthy and clinical populations [40,41,42,43]. However, this hypothesis about the role of a more immersive VR experience in diminishing psychological distress must be further elucidated through, e.g., RCTs. Interventions based on augmented reality influence, for instance, the ability to regulate emotions [36]. Moreover, exercising in an immersive virtual reality was found to be effective in diminishing symptoms of depression, anxiety, and stress [44]. However, less is known about the role of different levels of immersion resulting from the use of VR devices in influencing mental state. Jennett et al. [28] reported that full immersion results in a loss of the sense of time passing, a loss of awareness of the real world and involvement, resulting in the feeling of being immersed in the new experience. This sounds very similar to the construct of “flow” introduced by Csikszentmihalyi et al. [45], which was found to be related to subjective well-being [46]. It would be interesting (and potentially practical) to investigate which aspects of VR solutions contribute to immersion to the greatest extent.

We also believe that the causal relationship between immersion and mental well-being should be considered from a long-term perspective to establish its prolonged effects. If such effects are discovered, a more immersive VR experience might be considered a solution with the potential to support prevention and intervention in the context of mental health. It may also be a partial remedy for the negative effects resulting from the increasingly frequent and prolonged use of technology such as virtual reality. According to Slater et al. [47], immersion can be boosted via enhancement in display resolution, field of view, movement degrees of freedom, number of senses stimulated, ability to track and update user inputs, and the ability to isolate the user from stimuli in the real world. It would be beneficial to consider the role of different aspects of immersion in enhancing subjective well-being (positive emotions). Ermi and Mäyrä [48] indicated at least three elements of this phenomenon: sensory, challenge-based and imaginative immersion, which should be treated additively; together, they contribute to the overall impression of immersion. The first is related to the audiovisual setting. It can be modified, for example, by changing the intensity of individual experiences, and its occurrence is facilitated by, among other factors, the presence of loud and interesting sounds that effectively distract attention from external stimuli. Challenge-based immersion is based on a balance between the demands of the augmented reality environment and the user’s capabilities. Balance or lack thereof may occur at various levels, from motor to complex cognitive operations such as strategic planning or decision-making. The imaginative aspect of immersion is based on imaginative involvement in the presented narrative and emotional connection with the characters. We believe that some aspects of immersion enhance subjective well-being and lower the negative consequences of using this technology more than others; this knowledge could help guide the development and further modification of VR solutions.

### 4.1. Statement on the Constraints of Generality of These Findings

While our study provides valuable insights into the potential benefits of VR-induced immersion, certain constraints must be considered when generalizing the findings. The study’s participant pool, comprising 419 individuals aged 10 to 80, may exhibit diverse characteristics that could impact the applicability of the results across broader populations. Additionally, the specific design features of the experimental VR device, such as the multijoint arm and navigation algorithms, may influence the outcomes and limit generalizability to other VR platforms. Furthermore, the study of a Polish population using Polish adaptations of assessment tools introduces cultural and linguistic considerations that may affect the transferability of the results to different cultural contexts. Recognizing these constraints is essential for obtaining a nuanced understanding of the study’s implications and emphasizes the need for future research to explore the universality of immersive VR interventions in anxiety reduction. Moreover, the lack of information regarding respondents’ prior experience with VR or other augmented reality systems represents a significant limitation, as such experience could be a crucial variable influencing the state of immersion. Future studies should address this aspect to provide a more comprehensive understanding of the factors affecting VR-induced immersion. Additionally, the settings for the two levels of VR technology tested were not identical, which may have introduced subtle extraneous influences beyond the experimental manipulation and should also be considered in future research.

In this early-stage evaluation, including participants using wheelchairs allowed us to gather preliminary data on immersion, explore the system’s accessibility, and identify areas for further optimization to ensure equitable user experiences. However, to minimize potential risks or suboptimal experiences for individuals with disabilities, our initial evaluation intentionally limited broader participation from these populations. We acknowledge the importance of involving individuals with diverse abilities and plan to conduct follow-up studies to further validate and enhance the technology. This phased approach aligns with standard practices in developing inclusive and user-centered systems.

### 4.2. Practical Implications for VR Design and Research

Our findings suggest several key takeaways for VR designers, enthusiasts, and researchers. The results highlight the critical role of intuitive navigation systems and advanced design features in enhancing immersion, which, in turn, can positively influence psychological outcomes such as anxiety reduction. Sensory, challenge-based, and imaginative immersion appear to be pivotal in shaping user experiences, and incorporating these elements could significantly enhance the therapeutic potential of VR. Moreover, the accessibility demonstrated by the device for individuals using wheelchairs emphasizes the importance of inclusive design, ensuring equitable VR experiences for users with diverse physical capabilities. Future VR solutions should also account for users’ prior experience with VR and tailor immersion levels accordingly to optimize effectiveness. These insights provide a foundation for advancing VR technology to support mental well-being and broaden its application in diverse populations.

## 5. Conclusions

This study highlights the potential of an upgraded VR experience to reduce anxiety by enhancing immersion. Participants using the experimental device reported greater immersion and lower post-test anxiety compared to those using standard equipment, despite no initial group differences. Immersion partially mediated the relationship between group assignment and post-VR anxiety, even after controlling for sex, age, initial anxiety, and wheelchair use. Notably, women in the experimental group experienced higher immersion, while those in the control group exhibited greater anxiety symptoms. Age and initial anxiety predicted post-VR anxiety, but sex and wheelchair use had no independent effects.

These findings underscore the critical role of immersion in reducing anxiety symptoms and demonstrate the upgraded device’s effectiveness. The study accounted for 42% of the variance in post-VR anxiety, offering valuable insights for developing accessible, immersive VR solutions for mental health interventions.

## Data Availability

The data set associated with the paper is available under the following name: Data—The Role of Immersive Experience in Anxiety Reduction: Evidence from Virtual Reality Session, DOI:10.17632/h34mjxxhmt.1.

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
