# Peer review of "The Role of Immersive Experience in Anxiety Reduction: Evidence from Virtual Reality Sessions"

_brainsci, 2024, doi:10.3390/brainsci15010014_

Round 1
Reviewer 1 Report
Comments and Suggestions for Authors
The Role of Immersive Experience in Anxiety Reduction: Evidence From Virtual Reality Sessions
This study investigated the potential of a newly designed VR experience to alleviate anxiety, focusing on the mediating role of VR-induced immersion. Participants were randomly assigned to experimental and control groups based on predefined project criteria, including a random allocation to the wheelchair-using group. Both groups used goggles for virtual space navigation, with the experimental group employing a multi-joint arm mounted on an aluminum frame and special algorithms to navigate without controllers.
The results indicate that individuals using the new VR device demonstrated increased immersion and reduced post-test anxiety levels. These findings highlight the significance of immersion in enhancing psychological well-being, mitigating the negative effects of VR technology, and providing insights for the future development and refinement of VR solutions.
The study is methodologically well-structured and particularly interesting, aligning with the journal's objectives. However, several serious weaknesses and gaps emerge, which need to be addressed and justified by the authors:
There is no justification for using a population of healthy individuals instead of people with disabilities who actually use wheelchairs in their daily lives.
Regarding the two psychometric instruments, The State–Trait Anxiety Inventory and the Immersion Questionnaire, it would be helpful to include sample items. This would allow readers unfamiliar with these scales to gain a clearer understanding of what they evaluate.
In both the abstract and the discussion section, the authors state that the application improves participants' well-being. This is inaccurate, as only anxiety levels were measured. The concept of well-being is much broader and is typically assessed using different psychometric tools. This point needs to be clarified and corrected in both the abstract and the discussion.
Concerning the Virtual Reality environment, there is no information provided about the platform, system architecture, or other methodological details. Additionally, there is no description of what the immersive environment presented to participants or how this aligns with the study's objectives. Specifically, what exactly was the immersive environment, and how were participants expected to navigate it?
Author Response
For research article
Response to Reviewer 1 Comments |
||
1. Summary |
|
|
Thank you very much for taking the time to review this manuscript. Please find the detailed responses below and in track changes in the re-submitted files. |
||
2. Point-by-point response to Comments and Suggestions for Authors |
||
Comments 1: There is no justification for using a population of healthy individuals instead of people with disabilities who actually use wheelchairs in their daily lives. |
||
Response 1: We appreciate the reviewer’s comment and the importance of ensuring that VR technologies are inclusive for individuals with disabilities. We updated the text in the Material and Methods section in Wheelchair condition paragraph: “ In this study, we deliberately involved healthy individuals using wheelchairs to simulate the experience of disability. This approach allowed us to test the universality of the VR design in a controlled and ethical manner while minimizing variability from other factors such as pre-existing conditions or comorbidities”. We also updated the text at the end of the Discussion session and we added the last paragraph: “In this early-stage evaluation, including participants using wheelchairs allowed us to gather preliminary data on immersion, explore the system's accessibility, and identify areas for further optimization to ensure equitable user experiences. However, to minimize potential risks or suboptimal experiences for individuals with disabilities, our initial evaluation intentionally limited broader participation from these populations. We acknowledge the importance of involving individuals with diverse abilities and plan to conduct follow-up studies to further validate and enhance the technology. This phased approach aligns with standard practices in developing inclusive and user-centered systems” |
||
Comments 2: Regarding the two psychometric instruments, The State–Trait Anxiety Inventory and the Immersion Questionnaire, it would be helpful to include sample items. This would allow readers unfamiliar with these scales to gain a clearer understanding of what they evaluate. |
||
Response 2: Thank you for this recommendation. We have, accordingly, included sample items. Comment 3: In both the abstract and the discussion section, the authors state that the application improves participants' well-being. This is inaccurate, as only anxiety levels were measured. The concept of well-being is much broader and is typically assessed using different psychometric tools. This point needs to be clarified and corrected in both the abstract and the discussion. Response 3: Thank you for this comment. We have replaced the term psychological well-being with subjective well-being, as it is associated with positive emotions and aligns with the hedonic perspective. We modified the text in the Abstract and Discussion section as well as we got rid of the “psychological well-being” term from the keywords. Comment 4: Concerning the Virtual Reality environment, there is no information provided about the platform, system architecture, or other methodological details. Additionally, there is no description of what the immersive environment presented to participants or how this aligns with the study's objectives. Specifically, what exactly was the immersive environment, and how were participants expected to navigate it? Response 4: Thank you for this comment. In the procedure section there are few sentences which describe how the participants navigated in the environment: “In both cases, the participants navigated in a virtual space (VR) via goggles. The experimental group moved with goggles attached to a multijoint arm mounted on an aluminum frame. The experimental group was assessed in a test environment using specially developed algorithms for navigating in the virtual space without the use of controllers. The participants moved naturally within the real-world space up to the designated boundary, and the position of the multijoint arm mounted on the aluminum frame was tracked. Once the boundary was reached, movement in the virtual space was controlled through head movements, changes in body rotation, and tilts. Upon returning to the study area, the participants’ movements were again detected from their real-world position. On the other hand, the control group was assessed in a test environment without using algorithms, where movement was accomplished via controllers. The participants stood or sat still, and navigation within the virtual space was performed via a joystick on the controllers”. We also added a description of the VR environment in the same Procedure section, which reads as follows: “Both groups, experimental and control, navigated the same VR environment, which depicted an old room from before World War II. The room was furnished with antique furniture and paintings, resembling a museum or a historical setting” |

Reviewer 2 Report
Comments and Suggestions for Authors
This work investigates how different levels of immersive virtual reality technologies affect anxiety after VR sessions involving differing levels of physical mobility with and without wheelchairs. Listed below are some concerns I noted that the authors must consider addressing:
- It would have been nice to see a video demonstration of the experience from the users' perspectives to be able to clearly understand and see what the experience was like. While the authors attempt to discuss this in prose, a video demonstration would more clearly communicate the experience to the reader.
- Page 2: The authors state "Immersion is very important and, according to some researchers studying the virtual reality of computer games, crucial for the satisfaction derived from experiencing augmented reality [7]." - Missing "is crucial"
- Page 2: "While joysticks offer ease of use, learning-based" - should be "leaning-based"
- In the section "Aim of the Study", the authors attempt to explain the manipulations. I encourage the authors to add a section/subsection titled "Study design" where they detail that this was a between-subjects study, manipulating the type of VR technology used (traditional vs modern with connected arm). Furthermore wheelchair vs non-wheelchair is also a manipulation, making this a 2X2 factorial design with both VR technology level and mobility level being manipulated as between-subjects factors, each featuring 2 levels.
- While the authors attempt to analyze the data using a regression analyses with predictors of immersion, age, anxiety at T0, group, wheelchair level as predictors, an important aspect that was not really considered was VR/video game experience. Research shows that experience on these aspects can significantly determine how users feel post such experiences. For these reasons, I encourage the authors to add such demographic data to the model. Additionally, why were factorial 2 Way ANCOVAs not considered? If the objective of the work is to determine and compare how the level of VR technology affects anxiety, statistical tests of differences may be more appropriate than predictive analyses with regressions. I hence recommend that the authors add additional ANCOVA analyses to test if there were differences between the levels of VR technology discussed.
- A limitation of this work is that the study was not conducted in the same exact setting for both levels of VR technology tested in this work. This introduces a confound because differences observed need not be a result of the manipulation but could rather be a consequence of other extraneous influences like location being different. While I understand that ecological validity is being targeted in this work, I encourage the authors to add this point to the limitations sections where they discuss all limitations associated with the work conducted here and the study design adopted.
- Could differences in the trends observed with the experimental and control groups be a simple consequence of the demographic being different in terms of experience. I encourage the authors to comment on this based on data gathered or to mention this as a limitation of their work.
- In the discussion section, the authors state "Notably, in the experimental group, women experienced greater immersion than men did, whereas in the control group, women presented more increased anxiety symptoms in both assessments." - The second clause is non sequitur to the first clause. The first clause talks about women experiencing greater immersion than men whereas the second clause talks about anxiety.
- Using factorial ANOVA analyses, I also suggest that the authors conduct a manipulation check and report those results to determine of the two VR technologies differed in the levels of immersion induced.
-What also seems missing a discussion about VR designers, enthusiasts and researchers must take away from the findings obtained in this work. Along these lines, I encourage the authors to list a set of takeaways that readers must come away with when it comes to VR technology and anxiety that the authors have discovered.
- While the authors attempt to motivate the need to evaluate wheelchair vs non-wheelchair given differences in accessability levels across individuals, what seems missing is an explanation of why accessability differences is likely to have an effect on anxiety. Is there any prior work on this? If yes, they should be discussed. If not, the authors must more effectively articulate the motivation for exploring the wheelchair factor manipulated in this work with respect to anxiety.
- It would also be helpful for the authors to list out all the hypotheses as bulletted points along with the specific research questions that they want answered by this work in a section titled research questions and hypotheses. Adding and mentioning this content in this section will increase the readabilty of this work.
Overall, this is interesting work. My suggestion is to include analyses methods that may be more relevant to the aims stated.
Author Response
Response to Reviewer 2 Comments
Thank you very much for taking the time and the effort to review this manuscript. Detailed responses can be found below, as well as in the track changes within the resubmitted files.
Comments 1: It would have been nice to see a video demonstration of the experience from the users' perspectives to be able to clearly understand and see what the experience was like. While the authors attempt to discuss this in prose, a video demonstration would more clearly communicate the experience to the reader.
Response 1: That is a great recommendation; however, we do not have such material, and we did not obtain formal consent from the study participants for videotaping them. Therefore, we cannot provide this type of material for the study.
Comments 2: Page 2: The authors state "Immersion is very important and, according to some researchers studying the virtual reality of computer games, crucial for the satisfaction derived from experiencing augmented reality [7]." - Missing "is crucial"
- Page 2: "While joysticks offer ease of use, learning-based" - should be "leaning-based"
Response 2: Thank you. We corrected those mistakes in the text.
Comment 3: In the section "Aim of the Study", the authors attempt to explain the manipulations. I encourage the authors to add a section/subsection titled "Study design" where they detail that this was a between-subjects study, manipulating the type of VR technology used (traditional vs modern with connected arm). Furthermore wheelchair vs non-wheelchair is also a manipulation, making this a 2X2 factorial design with both VR technology level and mobility level being manipulated as between-subjects factors, each featuring 2 levels.
Response 3: Thank you, we have added the following subsection to the manuscript:
Study Design
This study employed a between-subjects experimental design to investigate the effects of two key variables on immersion and anxiety during virtual reality (VR) experiences. The study manipulated two independent variables: the level of VR technology and mobility condition, resulting in a 2x2 factorial design. Participants were randomly assigned to one of the four experimental conditions: (1) traditional VR + without wheelchair, (2) modern VR + without wheelchair, (3) traditional VR + with wheelchair, or (4) modern VR + with wheelchair. Each participant experienced only one combination of conditions, ensuring that both independent variables operated as between-subjects factors.
Comment 4: While the authors attempt to analyze the data using a regression analyses with predictors of immersion, age, anxiety at T0, group, wheelchair level as predictors, an important aspect that was not really considered was VR/video game experience. Research shows that experience on these aspects can significantly determine how users feel post such experiences. For these reasons, I encourage the authors to add such demographic data to the model.
Response 4: Thank you very much for this comment. We realize that this is very important variable. However we did not measured the experience therefore we added this aspect in the paragraph about the limitations and future steps paragraph in the Discussion section: “Moreover, the lack of information regarding respondents' prior experience with VR or other augmented reality systems represents a significant limitation, as such experience could be a crucial variable influencing the state of immersion. Future studies should address this aspect to provide a more comprehensive understanding of the factors affecting VR-induced immersion.”
Comment 4a: Additionally, why were factorial 2 Way ANCOVAs not considered? If the objective of the work is to determine and compare how the level of VR technology affects anxiety, statistical tests of differences may be more appropriate than predictive analyses with regressions. I hence recommend that the authors add additional ANCOVA analyses to test if there were differences between the levels of VR technology discussed.
Response 4a: Thank you for your thoughtful comment regarding the use of factorial ANCOVA analyses and the inclusion of additional demographic data. We appreciate the opportunity to clarify our methodological choice:
We agree that factorial ANCOVA is a suitable method for analyzing group differences while controlling for covariates. However, we opted for a linear regression model because it allows for a more nuanced exploration of the relationship between the independent variables (e.g., VR technology, mobility condition, immersion) and the change in anxiety between pre- (T0) and post-test (T1). This approach aligns with the longitudinal nature of our study, as it focuses on how the predictors influence the change in the dependent variable over time rather than simply comparing group means at a single time point.
While ANCOVA and regression are mathematically related and often yield similar results when covariates are included, regression models provide greater flexibility for modeling continuous outcomes and interaction effects in longitudinal designs. Additionally, our regression model allowed us to incorporate predictors such as baseline anxiety and immersion, which are key variables of interest in understanding the psychological processes underlying the effects of VR technology.
We acknowledge that prior VR or video game experience could influence participants' responses to the VR sessions. Unfortunately, we did not collect this specific demographic data in the current study. We recognize this as a limitation and have added a statement to the Limitations section, highlighting the importance of including this variable in future research.
We hope this explanation clarifies our methodological choices and provides a rationale for why regression was used in this longitudinal study. We believe that the regression approach is well-suited to our objectives and effectively captures the dynamic changes in anxiety over time.
Comments 5: A limitation of this work is that the study was not conducted in the same exact setting for both levels of VR technology tested in this work. This introduces a confound because differences observed need not be a result of the manipulation but could rather be a consequence of other extraneous influences like location being different. While I understand that ecological validity is being targeted in this work, I encourage the authors to add this point to the limitations sections where they discuss all limitations associated with the work conducted here and the study design adopted.
Response 5: Thank you for this comment. We added the limitation in the end of the Discussion section: “Additionally, the settings for the two levels of VR technology tested were not identical, which may have introduced subtle extraneous influences beyond the experimental manipulation and should also be considered in future research”
Comment 6: Could differences in the trends observed with the experimental and control groups be a simple consequence of the demographic being different in terms of experience. I encourage the authors to comment on this based on data gathered or to mention this as a limitation of their work.
Response 6: Thank you for that comment. We have added the experience as the limitation and future steps.
Comment 7: In the discussion section, the authors state "Notably, in the experimental group, women experienced greater immersion than men did, whereas in the control group, women presented more increased anxiety symptoms in both assessments." - The second clause is non sequitur to the first clause. The first clause talks about women experiencing greater immersion than men whereas the second clause talks about anxiety.
Response 7: Thank you for pointing this out. We changed the sentence: “Notably, in the experimental group, women experienced greater immersion than men, while in the control group, women showed higher anxiety symptoms in both assessments, highlighting potential gender differences in the response to VR-induced immersion and anxiety.”
Comment 8: Using factorial ANOVA analyses, I also suggest that the authors conduct a manipulation check and report those results to determine of the two VR technologies differed in the levels of immersion induced.
Response 8: Thank you for your suggestion to include a manipulation check to determine whether the two VR technologies differed in the levels of immersion induced. We appreciate your recommendation and would like to clarify that this analysis has already been conducted and is presented in the manuscript.
Specifically, the differences in immersion levels between the experimental groups (traditional VR vs. modern VR) are reported in Table 2 under the section "Results." As shown, the results indicate significant differences in immersion scores between the two VR technologies, with the modern VR setup producing higher levels of immersion compared to the traditional VR technology (p < .05).
We hope this addresses your concern and clarifies that a manipulation check was indeed performed and documented in the manuscript. Thank you for your valuable feedback, which ensures clarity and thoroughness in presenting our findings.
Comment 9: What also seems missing a discussion about VR designers, enthusiasts and researchers must take away from the findings obtained in this work. Along these lines, I encourage the authors to list a set of takeaways that readers must come away with when it comes to VR technology and anxiety that the authors have discovered.
Response 9: Thank you very much for this valuable comment. We added the paragraph at the end of the Discussion section:
“Practical Implications for VR Design and Research
Our findings suggest several key takeaways for VR designers, enthusiasts, and researchers. The results highlight the critical role of intuitive navigation systems and advanced design features in enhancing immersion, which, in turn, can positively influence psychological outcomes such as anxiety reduction. Sensory, challenge-based, and imaginative immersion appear to be pivotal in shaping user experiences, and incorporating these elements could significantly enhance the therapeutic potential of VR. Moreover, the accessibility demonstrated by the device for individuals using wheelchairs emphasizes the importance of inclusive design, ensuring equitable VR experiences for users with diverse physical capabilities. Future VR solutions should also account for users’ prior experience with VR and tailor immersion levels accordingly to optimize effectiveness. These insights provide a foundation for advancing VR technology to support mental well-being and broaden its application in diverse populations”
Comment 10: While the authors attempt to motivate the need to evaluate wheelchair vs non-wheelchair given differences in accessability levels across individuals, what seems missing is an explanation of why accessability differences is likely to have an effect on anxiety. Is there any prior work on this? If yes, they should be discussed. If not, the authors must more effectively articulate the motivation for exploring the wheelchair factor manipulated in this work with respect to anxiety.
Response 10: Thank you for this comment, we added the text to the third paragraph of the Discussion section: “Building on this, we believe that future research should further explore how factors such as mobility, accessibility, and inclusivity in VR environments directly influence mental states like anxiety and emotional well-being, particularly in populations with specific physical or psychological vulnerabilities”
Comment 11: It would also be helpful for the authors to list out all the hypotheses as bulletted points along with the specific research questions that they want answered by this work in a section titled research questions and hypotheses. Adding and mentioning this content in this section will increase the readabilty of this work.
Response 11: Thank you. To enhance clarity and readability, we have outlined below the specific research questions and hypotheses that guided this study in the introduction section:
Research questions and hypotheses
This study explores the following research questions, along with the corresponding hypotheses:
- Does the use of modern VR technology (with a multijoint arm and natural navigation) result in higher levels of immersion compared to traditional VR technology (with controllers)?
H1: Participants using modern VR technology will report significantly higher immersion levels compared to those using traditional VR technology.
- How does the type of VR technology influence changes in anxiety levels before and after the VR session?
H2: Modern VR technology will lead to greater reductions in post-session anxiety compared to traditional VR technology.
- Do mobility conditions (with or without wheelchair) affect immersion levels and anxiety reduction in VR environments?
H3: Mobility condition (with or without wheelchair) will not significantly impact immersion levels or anxiety reduction.
- Does immersion mediate the relationship between VR technology (group) and anxiety levels after the VR session (T1), when controlling for variables such as sex, age, baseline anxiety (T0), and mobility condition?
H4: Immersion will act as a mediator between VR technology (group) and post-session anxiety (T1), such that higher immersion levels associated with modern VR technology will explain greater reductions in anxiety.